# *Sekentei* as a Socio-Cultural Determinant of Cognitive Function among Older Japanese People: Findings from the NEIGE Study

**DOI:** 10.3390/ijerph17124480

**Published:** 2020-06-22

**Authors:** Hiroshi Murayama, Shigeru Inoue, Takeo Fujiwara, Naoki Fukui, Yuichi Yokoyama, Yugo Shobugawa

**Affiliations:** 1Research Team for Social participation and Community Health, Tokyo Metropolitan Institute of Gerontology, Tokyo 173-0015, Japan; 2Department of Preventive Medicine and Public Health, Tokyo Medical University, Tokyo 160-8402, Japan; inoue@tokyo-med.ac.jp; 3Department of Global Health Promotion, Tokyo Medical and Dental University, Tokyo 113-8510, Japan; fujiwara.hlth@tmd.ac.jp; 4Department of Psychiatry, Niigata University Graduate School of Medical and Dental Sciences, Niigata 951-8510, Japan; fukui@med.niigata-u.ac.jp (N.F.); yokoyama@med.niigata-u.ac.jp (Y.Y.); 5Division of International Medicine, Niigata University Graduate School of Medical and Dental Sciences, Niigata 951-8510, Japan; yugo@med.niigata-u.ac.jp

**Keywords:** cognitive function, *sekentei*, social appearance, social norms, older people, Japan

## Abstract

*Sekentei* (social appearance) is a Japanese concept that describes a person’s sense of implicit societal pressure to conform to social norms. However, evidence of a relationship between *sekentei* and health outcomes is sparse. This study examined the association between *sekentei* and cognitive function among community-dwelling older Japanese people. Baseline data were obtained from the Neuron to Environmental Impact across Generations (NEIGE) study conducted in 2017; 526 randomly sampled community-dwelling individuals aged 65–84 years living in Tokamachi, Niigata Prefecture, Japan were analyzed. The 12-item Sekentei Scale was used to assess *sekentei*. Cognitive function levels were evaluated with the Japanese version of Mini-Mental State Examination (MMSE-J; ranging from 0–30). Approximately 10% and 25% had cognitive decline and mild cognitive impairment, respectively (MMSE-J scores of ≤23 and 24–26, respectively). Multinomial logistic regression analysis showed that both high and low levels of *sekentei* were associated with lower cognitive function, particularly mild cognitive impairment, after adjusting for sociodemographic factors, health behaviors, health conditions, and genetic factors. The current findings suggest that a moderate level of *sekentei* consciousness is beneficial for cognitive health, and that *sekentei* could be an important socio-cultural factor affecting cognitive function.

## 1. Introduction

The global burden of dementia is increasing worldwide, and the development of measures for dementia is an increasingly important global public health issue [1,2]. In Japan, which is one of the most rapidly aging nations in the world, the number of older people with dementia is predicted to exceed 7 million by 2025 and 11 million by 2060 (approximately 20% and 33% of the older population, respectively) [3].

The risk factors for cognitive decline have been explored in many previous studies: medical (for example, diabetes and hypertension), nutritional (e.g., fruit and vegetable intake), socioeconomic (e.g., education and occupation), behavioral (e.g., smoking, physical activity, and cognitive engagement), and genetic (e.g., apolipoprotein E (APOE)) factors have been reported in systematic reviews [4,5]. For example, evidence for physical activity as a means to decrease risk of cognitive decline was shown both in observational studies and in a randomized controlled trial [4]. 

In discussions about the social determinants of health, the importance of socio-cultural factors such as social norms has been proposed [6]. Social norms are defined as unwritten rules about how to behave in a particular social group or culture [7]. People interpret the attitudes and behaviors of others around them as information about what should be done in a given situation, and often act accordingly [8]. Indeed, health behaviors can be influenced by social norms: physical exercise [9], smoking [10], and drug use [11]. However, to the best of our knowledge, no previous studies have examined the socio-cultural determinants of cognitive decline, including social norms.

The concept of *sekentei*, which refers to social appearance or sensitivity regarding one’s reputation [12], is considered to be an important socio-cultural factor determining behavioral principles of people in Japan [12]. *Sekentei* is specific to Japanese culture [13] and implicitly applies pressure to conform to social norms. The term is used to refer to an individual’s concerns about having to meet the standards of socially acceptable manners, which are judged by others [12]. Thus, *sekentei* can also be defined as an awareness that a person feels about others observing and evaluating their behaviors [14].

Several studies have found that people with a high level of *sekentei* consciousness tend to avoid the use of care services [15,16]. In addition, a relationship between *sekentei* and health behaviors has been reported: higher *sekentei* consciousness was associated with lower physical activity among older Japanese adults [17]. However, evidence regarding the relationship between *sekentei* and health outcomes is still lacking. Because *sekentei* is a fundamental socio-cultural factor determining people’s attitudes and behaviors in Japan, it may be linked to cognitive function. 

In the current study, we examined the association between *sekentei* and cognitive function among community-dwelling older people in Japan. Focusing on *sekentei* as a socio-cultural factor may contribute to a deeper understanding of the determinants of cognitive decline.

## 2. Materials and Methods 

### 2.1. Study Sample and Data Collection

Data were obtained from the baseline survey of the Neuron to Environmental Impact across Generations (NEIGE) study in September and October 2017. The NEIGE study was conducted in Tokamachi, Niigata Prefecture, Japan. Tokamachi is a rural city in the southernmost region of Niigata Prefecture, which is approximately 180 km northwest of Tokyo. As of 1 July 2017, the population of the city was 54,515 (26,560 men and 27,955 women), with 20,089 people aged ≥65 years (the proportion of older adults was 36.9%).

The baseline survey of the NEIGE study included randomly recruited community-dwelling older adults aged 65 to 84 years, who lived in Tokamachi and were not a recipient of long-term care insurance (that is, independent living). We stratified the residents of Tokamachi into four groups according to their age (65–74 years and 75–84 years) and their area of residence within the town (downtown area and mountainside area). Consequently, from a total of 15,792 eligible participants (average age 73.6, 47.0% men), 1524 were randomly sampled from these four groups. 

After excluding those with long-term care certification, admitted to hospital, and nursing home residents, 1346 people eligible for research were selected, and they received the recruitment brochure to participate in the NEIGE study by mail. Finally, we obtained the agreement to participate in the NEIGE study from 527 people (participation rate: 39.2%). Further information on sampling and participant demographics is provided in a previous paper [18]. 

Participants of the NEIGE study were interviewed face-to-face by trained interviewers to collect comprehensive information about their physical, mental, cognitive, and social functions. The study protocol was reviewed and approved by the Ethical Committees of Niigata University on November 25, 2016 (approval number: 2666). All participants gave written consent to participate in this study.

### 2.2. Measures

#### 2.2.1. Cognitive Function

Participants’ cognitive function was assessed using the Mini-Mental State Examination, Japanese version (MMSE-J) comprising 30 items [19]. The MMSE is widely used as a brief screening test for dementia and is a measure of global cognitive ability. MMSE-J scores range from 0 to 30, and lower scores indicate poorer global cognitive ability. For data collection, the MMSE was administered by well-trained staff. This study adopted two cut-off points: ≤23 and ≤26. The former cut-off threshold is widely used to detect cognitive decline (a score of ≤23 indicated cognitive decline) [19,20]. The latter cut-off score has also been used in some studies to distinguish people with mild cognitive impairment (MCI) and healthy individuals (a score of ≤26 indicates MCI) [21,22]. Therefore, we used three categories in the analysis: ≤23 (cognitive decline), 24–26 (MCI), and ≥27 (cognitively healthy). 

#### 2.2.2. Sekentei

We used the Sekentei Scale, which comprises 12 items rated on a 5-point Likert scale (1 = strongly disagree, 2 = disagree, 3 = neither, 4 = agree, or 5 = strongly agree) [16,23]. This scale captured respondents’ *sekentei* levels. The scale includes items such as, “I tend to adjust my actions according to the behaviors of people around me”, “I am rather unconcerned about gossip and the way I appear to others (reverse item)”, “I avoid behavior that people laugh at”, and, “I would definitely return the favor if I were cared for by or received a gift from others”. The validity (i.e., content validity and construct validity) and reliability (i.e., internal consistency reliability) of this scale have been confirmed [16]. The score range of the scale is 12–60. A higher score indicates a greater sense of *sekentei*. Cronbach’s alpha in the current study was 0.61. Because there was no previous research examining the relationship between Sekentei Scores and cognitive function, it was unclear whether the relation was linear. Therefore, in the analysis, we divided the scores into quintiles (≤36, 37–39, 40–41, 42–44, and ≥45). 

#### 2.2.3. Covariates

Sociodemographic factors, health behaviors, health conditions, and genetic factors were used as covariates because these factors were considered to be confounders of the association between *sekentei* and cognitive function. Sociodemographic factors included age, gender, years of residence in the area, marital status (married or not married), living alone (yes or no), current working status (working or not working), years of education (≤9 years or ≥10 years), and subjective financial stability (1 = poor, 2 = somewhat poor, 3 = normal, 4 = somewhat affluent, or 5 = affluent). Information on age and gender was obtained from the residential registry. 

Smoking status, daily walking time (1 = <30 min, 2 = 30–59 min, 3 = 60–89 min, or 4 = ≥90 min), and body mass index were included as health behaviors. Body mass index was calculated from actual measurements of height and weight (kg/m^2^), and participants were classified into three categories: underweight (<18.5), normal weight (18.5–24.9), and overweight (≥25.0).

Health conditions included comorbidities and depressive mood. Medical interviews by a doctor or registered nurse were conducted to retrieve information regarding comorbidities. Six diagnosed diseases (cancer, hypertension, cardiovascular disease, cerebrovascular disease, dyslipidemia, and diabetes mellitus) were used, and participants were categorized into three groups based on the number of comorbid diseases, 0, 1 and ≥2. Depressive mood was assessed using the Geriatric Depression Scale (GDS) short-form, which comprises 15 items with dichotomized responses [24,25]. We used a cut-off point of ≥6, which indicated a depressive mood [25].

To examine genetic factors, APOE genotyping was performed. Genomic DNA was obtained from whole-blood samples to determine APOE genotypes using a standard polymerase chain reaction technique. The technicians handling the coded DNA specimens were blinded to the diagnosis. The categorical variable of APOE was classified by genotype as ε2ε2, ε2ε3, ε2ε4, ε3ε3, ε3ε4, and ε4ε4 [26].

### 2.3. Statistical Analyses

First, we compared the characteristics of the participants by their level of cognitive function based on MMSE-J scores. Second, we examined the association between *sekentei* and cognitive function using a multinomial logistic regression analysis. Because we divided cognitive function into three categories (i.e., cognitive decline, MCI, and cognitively healthy), we adopted a multinomial logistic regression model to understand the detailed relationship between *sekentei* and cognitive function. We used a three-step modeling strategy. Age, gender, and Sekentei Scale score were included in Model 1. Model 2 included years of residence, marital status, living alone, current working status, years of education, and subjective financial stability in addition to Model 1. Finally, we added health behaviors (smoking status, daily walking time, and body mass index), health conditions (comorbidities and depressive mood), and genetic factors (APOE genotype) in Model 3. The results are shown as odds ratios (ORs) with 95% confidence intervals (CIs). The analyses were performed using the IBM SPSS 23 (IBM Corp., Armonk, NY, USA). 

## 3. Results

Of the 527 participants, 526 completed the MMSE-J and were included in the analyses. Table 1 shows the participants’ characteristics. The average age was 73.5 years old (standard deviation: 5.6) and 47.3% were men, which was similar to the characteristics of the target population. Regarding socioeconomic status, 38.2% had received less than nine years of education, and 23.9% considered themselves poor. A total of 65.6% of participants had more than one chronic disease, and 16.5% had a score of ≥6 on the GDS. In terms of cognitive function, the proportions of those with MMSE-J scores of ≤23, 24–26, and ≥27 were 9.9%, 23.3%, and 66.7%, respectively. The average Sekentei Scale score was 41.6 (standard deviation: 5.0, median: 42).

Table 2 represents the participants’ characteristics by cognitive function. Those with MMSE-J ≤23 (cognitive decline) were older and had lived longer in the area. Moreover, they tended to not be working, to have fewer years of education, to be poorer, and to have a depressive mood. There were no significant differences in APOE genotype (that is, the proportion of those having at least one copy of ε4) and Sekentei Scale scores among the three groups. The proportion of men was higher in the MCI group (MMSE-J scores of 24–26) than in the other groups. 

Table 3 indicates the association between *sekentei* and cognitive function. Those with the lowest and highest levels of *sekentei* (that is, the first and fifth quintiles) were more likely to have MCI, compared with those with moderate-level (that is, the fourth quintile) Sekentei Scale scores (ORs (95% CIs) were 2.44 (1.21–4.92) for the first quintile and 1.98 (1.07–3.65) for the fifth quintile), when adjusted for age and gender in Model 1. This association remained significant when the results were adjusted for sociodemographic factors, health behaviors, health conditions, and genetic factors in Models 2 and 3 (e.g., 2.37 (1.13–4.98) for the first quintile, and 2.16 (1.13–4.12) for the fifth quintile in Model 3). Although the results did not reach statistical significance, this trend was also observed for cognitive decline (e.g., 1.45 (0.42–4.96) for the first quintile and 1.72 (0.69–4.33) for the fifth quintile in Model 3).

We added interaction terms between *sekentei* and sociodemographic factors in Model 3, but no significant interaction was found (data not shown in the table). Thus, the effect of *sekentei* consciousness might not vary according to individual sociodemographic characteristics. 

As a sensitivity analysis, we divided the Sekentei Scale score into sextiles instead of quintiles and conducted multinomial logistic regression analysis. We found a similar trend in the association between *sekentei* and cognitive function: higher and lower *sekentei* levels were related to lower cognitive function, particularly MCI. 

## 4. Discussion

The current study examined the relationship between *sekentei* and cognitive function, using data from the NEIGE study, which comprises randomly sampled community-dwelling older Japanese people. We found that both higher and lower *sekentei* levels were negatively associated with lower cognitive function, particularly MCI. *Sekentei* is a behavioral principle in Japanese culture [12,13], and a previous study reported a link between *sekentei* and physical inactivity [17]. However, to the best of our knowledge, no previous study has explored the relationship between *sekentei* and health outcomes, including cognitive function. The current findings contribute to the understanding of the cultural determinants of cognitive decline. 

The highest level of *sekentei* (that is, the highest quintile) was associated with MCI. A higher level of *sekentei* indicates greater sensitivity regarding one’s reputation [14]; therefore, people with higher *sekentei* consciousness may exhibit more neurotic personality traits. A systematic review reported that higher neuroticism was associated with greater risks of dementia and mild cognitive impairment [27] because neuroticism was negatively associated with higher intellectual ability [28], which was protective against dementia owing to brain reserve [29]. Therefore, an excessively high level of *sekentei* consciousness might not be beneficial for cognitive functioning in old age. 

At the same time, the lowest *sekentei* level (that is, the lowest quintile) was also correlated with MCI. People with lower *sekentei* consciousness are likely to worry little about how they are coming across to others (that is, their reputation among others). Although these individuals may seem to be carefree, this attitude is not always beneficial for health. One previous study reported a beneficial role of brief stress on the hippocampus [30]. Moreover, cognitive stimulation in daily life might have a favorable effect against cognitive decline. A systematic review revealed that cognitive function in older adults could be improved through intellectual stimulation, such as cognitive leisure activity interventions [31]. Another study reported that having heterogeneous social relationships can prevent cognitive decline by obtaining various types of novel information and inspiring ideas among older people [32]. Therefore, it could be concluded that an excessively low level of *sekentei* consciousness was also detrimental for cognitive function.

The associations of *sekentei* with cognitive decline and MCI were U-shaped, but the association of high and low *sekentei* levels with MCI was stronger than that with cognitive decline. This result might have been caused by the following factors. First, the reliability of responses to the Sekentei Scale among people with cognitive decline might be low because of the misclassification of responses on the Sekentei Scale. Second, there might have been unmeasured confounding effects on the association, such as neighborhood environment factors and social relationships with residents. This possibility should be investigated in future research for further understanding of the relationship. 

Several limitations involved in the current study should be considered. First, selection bias may have occurred because our study sample was likely to have contained particularly healthy individuals. Despite the random sampling of participants, people with cognitive decline or dementia would be expected to be less likely to participate in the survey. Second, because of the cross-sectional nature of this study, causal relationships could not be determined; longitudinal data are required, which should be collected in a future study. Previous meta-analyses have defined social cognition as the understanding of another person’s knowledge, beliefs, emotions, and intentions, and the ability to use that understanding to navigate social situations. This function was impaired both in frontotemporal and Alzheimer’s disease dementia [33,34]. Thus, cognitive decline or MCI may cause impaired social cognition, resulting in extreme responses on the Sekentei Scale (that is, low and high scores). Third, Cronbach’s alpha of the Sekentei Scale in this study was 0.61, which indicated that the internal consistency reliability was not high. The scale was originally developed based on a survey for young, middle-aged, and older people [16,23]. Because the sample of the current study was limited to the older population, we could not obtain enough internal consistency of the scale. Fourth, this research focused on a single geographical location. Caution should be used when generalizing the findings. Finally, the participants in this study were 65–84 years old; therefore, further examination is necessary to determine whether our findings can be applied to other age groups, such as those ≥85 years old. 

## 5. Conclusions

Using the baseline data of the NEIGE study, we explored the association between *sekentei,* which is a normative awareness reflecting Japanese behavioral principles, and cognitive function among community-dwelling older Japanese people. The association was found to be U-shaped: both high and low levels of *sekentei* were associated with lower cognitive function, particularly MCI, after adjusting for sociodemographic factors, health behaviors, health conditions, and genetic factors. Thus, a moderate level of *sekentei* consciousness was found to be beneficial to maintaining cognitive health. The findings show that *sekentei* may be an important socio-cultural factor that affects cognitive decline and suggest the importance of a culturally appropriate approach to prevent cognitive decline in the community.

## Figures and Tables

**Table 1 ijerph-17-04480-t001:** Participants’ characteristics (*N* = 526).

Variable	Category	Mean (SD)	*n*	%
Age (year)		73.5 (5.6)		
Gender	Men		249	47.3%
	Women		277	52.7%
Years of residence in the area		53.9 (17.6)		
Marital status	Married		423	80.4%
	Unmarried		103	19.6%
Living alone	Yes		47	8.9%
	No		479	91.1%
Current working status	Working		217	41.3%
	Not working		309	58.7%
Years of education	≤9		201	38.2%
	≥10		325	61.8%
Subjective financial stability	Very poor		26	4.9%
	Poor		100	19.0%
	Normal		321	61.0%
	Affluent		68	12.9%
	Very affluent		11	2.1%
Current smoking status	Smoking		47	8.9%
	Not smoking		479	91.1%
Daily walking time (minutes)	<30		109	20.7%
	30–59		190	36.1%
	60–89		95	18.1%
	≥90		132	25.1%
Body mass index (kg/m^2^)	Overweight (≥25.0)		95	18.1%
	Normal weight (18.5–24.9)		389	74.0%
	Underweight (<18.5)		42	8.0%
Comorbidities	0		181	34.4%
	1		193	36.7%
	≥2		152	28.9%
Depressive mood (GDS)	≥6		87	16.5%
	≤5		435	82.7%
	Missing		4	0.8%
APOE genotype	ε2ε2		1	0.2%
	ε2ε3		45	8.6%
	ε2ε4		5	1.0%
	ε3ε3		401	76.2%
	ε3ε4		66	12.5%
	ε4ε4		7	1.3%
	Missing		1	0.2%
Cognitive function (MMSE-J)	≤23		52	9.9%
	24–26		123	23.3%
	≥27		351	66.7%
Sekentei Scale score (possible range: 12–60)		41.6 (5.0)		

APOE: apolipoprotein E. GDS: Geriatric Depression Scale. MMSE-J: Mini-Mental State Examination, Japanese version. SD: standard deviation.

**Table 2 ijerph-17-04480-t002:** Comparison of participants’ characteristics by cognitive function.

Variable	Category	MMSE-J, ≤23			MMSE-J, 24–26			MMSE-J, ≥27			*p*-Value
		Mean (SD)	*n*	%	Mean (SD)	*n*		Mean (SD)	*n*	%	
Age (year)		77.7 (5.3)			74.5 (5.5)			72.5 (5.3)			<0.001 ^a^
Gender	Men		25	48.1%		71	57.7%		153	43.6%	0.026 ^b^
	Women		27	51.9%		52	42.3%		198	56.4%	
Years of residence in the area		57.7 (18.5)			56.8 (16.8)			52.3 (17.5)			0.013 ^a^
Marital status	Married		37	71.2%		102	82.9%		284	80.9%	0.185 ^b^
	Unmarried		15	28.8%		21	17.1%		67	19.1%	
Living alone	Yes		8	15.4%		9	7.3%		30	8.5%	0.210 ^b^
	No		44	84.6%		114	92.7%		321	91.5%	
Current working status	Working		12	23.1%		49	39.8%		156	44.4%	0.013 ^b^
	Not working		40	76.9%		74	60.2%		195	55.6%	
Years of education	≤9		36	69.2%		54	43.9%		111	31.6%	<0.001 ^b^
	≥10		16	30.8%		69	56.1%		240	68.4%	
Subjective financial stability	Very poor		4	7.7%		8	6.5%		14	4.0%	0.026 ^c^
	Poor		8	15.4%		31	25.2%		61	17.4%	
	Normal		35	67.3%		71	57.7%		215	61.3%	
	Affluent		5	9.6%		10	8.1%		53	15.1%	
	Very affluent		0	0.0%		3	2.4%		8	2.3%	
Current smoking status	Smoking		3	5.8%		17	13.8%		27	7.7%	0.086 ^b^
	Not smoking		49	94.2%		106	86.2%		324	92.3%	
Daily walking time (minutes)	< 30		13	25.0%		21	17.1%		75	21.4%	0.066 ^c^
	30–59		18	34.6%		38	30.9%		134	38.2%	
	60–89		10	19.2%		24	19.5%		61	17.4%	
	≥90		11	21.2%		40	32.5%		81	23.1%	
Body mass index (kg/m^2^)	Overweight (≥25.0)		10	19.2%		24	19.5%		61	17.4%	0.754 ^c^
	Normal weight (18.5–24.9)		36	69.2%		91	74.0%		262	74.6%	
	Underweight (<18.5)		6	11.5%		8	6.5%		28	8.0%	
Comorbidities	0		12	23.1%		43	35.0%		126	35.9%	0.837 ^c^
	1		28	53.8%		44	35.8%		121	34.5%	
	≥2		12	23.1%		36	29.3%		104	29.6%	
Depressive mood (GDS)	≥6		15	30.0%		24	19.8%		48	13.7%	0.008 ^b^
	≤5		35	70.0%		97	80.2%		303	86.3%	
APOE genotype	Has at least one copy of ε4 (ε2ε4/ε3ε4/ε4ε4)		12	23.1%		15	12.2%		51	14.6%	0.175 ^b^
	Has no copy of ε4 (ε2ε2/ε2ε3/ε3ε3)		40	76.9%		108	87.8%		299	85.4%	
Sekentei Scale score (possible range: 12–60)		42.7 (4.7)			41.7 (5.4)			41.3 (4.8)			0.176 ^a^

APOE: apolipoprotein E. GDS: Geriatric Depression Scale. MMSE-J: Mini-Mental State Examination, Japanese version. SD: standard deviation. Missing values were removed. ^a^ Analysis of variance. ^b^ Chi-square test. ^c^ Kruskal–Wallis test.

**Table 3 ijerph-17-04480-t003:** Association between *sekentei* and cognitive function: a multinomial logistic regression analysis.

Variable	Category	Model 1		Model 2		Model 3	
		MMSE-J, ≤23	MMSE-J, 24–26	MMSE-J, ≤23	MMSE-J, 24–26	MMSE-J, ≤23	MMSE-J, 24–26
		OR (95% CI)	OR (95% CI)	OR (95% CI)	OR (95% CI)	OR (95% CI)	OR (95% CI)
Sekentei Scale score	1st quintile (lowest)	1.17 (0.38–3.65)	2.44 (1.21–4.92)	1.35 (0.42–4.37)	2.50 (1.22–5.13)	1.45 (0.42–4.96)	2.37 (1.13–4.98)
	2nd quintile	0.89 (0.33–2.44)	1.11 (0.54–2.29)	0.96 (0.34–2.75)	1.11 (0.53–2.31)	1.16 (0.39–3.46)	1.07 (0.51–2.27)
	3rd quintile	1.23 (0.49–3.11)	1.57 (0.79–3.11)	1.45 (0.55–3.80)	1.63 (0.81–3.27)	1.49 (0.53–4.20)	1.60 (0.78–3.28)
	4th quintile	1.00	1.00	1.00	1.00	1.00	1.00
	5th quintile (highest)	1.36 (0.60–3.08)	1.98 (1.07–3.65)	1.55 (0.66–3.63)	2.08 (1.11–3.89)	1.72 (0.69–4.33)	2.16 (1.13–4.12)
Age	(Every 1-year increase)	1.19 (1.12–1.26)	1.08 (1.03–1.12)	1.15 (1.08–1.23)	1.06 (1.01–1.11)	1.16 (1.07–1.24)	1.07 (1.02–1.12)
Gender	Women	0.81 (0.44–1.50)	0.56 (0.36–0.85)	0.50 (0.24–1.02)	0.51 (0.31–0.83)	0.54 (0.24–1.19)	0.53 (0.31–0.89)
Years of residence in the area	(Every 10-year increase)			0.96 (0.79–1.17)	1.03 (0.90–1.19)	1.03 (0.83–1.27)	1.04 (0.90–1.20)
Marital status	Married			1.03 (0.40–2.67)	1.18 (0.56–2.47)	1.20 (0.44–3.29)	1.13 (0.53–2.43)
Living alone	Yes			1.47 (0.46–4.69)	0.96 (0.35–2.58)	1.59 (0.45–5.56)	0.84 (0.30–2.40)
Current working status	Working			0.53 (0.25–1.12)	0.87 (0.55–1.36)	0.46 (0.21–1.02)	0.80 (0.50–1.28)
Years of education	≤9 years			3.51 (1.74–7.10)	1.63 (1.01–2.63)	3.05 (1.46–6.39)	1.50 (0.92–2.45)
Subjective financial stability	(Poorer)			0.85 (0.55–1.30)	0.72 (0.54–0.96)	0.96 (0.60–1.54)	0.77 (0.57–1.04)
Current smoking status	Smoking					1.01 (0.25–4.06)	1.82 (0.87–3.82)
Daily walking time	(Longer ^a^)					1.19 (0.86–1.65)	1.32 (1.07–1.64)
Body mass index (kg/m^2^)	Overweight (≥25.0)					1.29 (0.53–3.11)	1.23 (0.69–2.20)
	Underweight (<18.5)					1.38 (0.44–4.27)	0.84 (0.34–2.05)
Comorbidities	1					2.61 (1.12–6.11)	0.99 (0.58–1.68)
	≥2					0.95 (0.35–2.58)	0.77 (0.43–1.37)
Depressive mood (GDS)	≥6					2.63 (1.16–5.96)	1.76 (0.97–3.21)
APOE genotype	ε4 (ε2ε4/ε3ε4/ε4ε4)					2.05 (0.89–4.74)	0.78 (0.40–1.51)

APOE: apolipoprotein E. CI: confidence interval. GDS: Geriatric Depression Scale. MMSE-J: Mini-Mental State Examination, Japanese version. OR: odds ratio. ^a^ The OR and 95% CI of daily walking time were calculated based on the response category (“1 = < 30 min”, “2 = 30–59 min”, “3 = 60–89 min”, or “4 = ≥ 90 min”).

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
