# Peer review of "Sekentei as a Socio-Cultural Determinant of Cognitive Function among Older Japanese People: Findings from the NEIGE Study"

_ijerph, 2020, doi:10.3390/ijerph17124480_

Round 1

Reviewer 1 Report

This is an interesting, novel, and generally well-written paper assessing associations between sekentei and cognitive function. The overall logic, structure, presentation, methods, and conclusions are good. Specific comments below.

Introduction:

This section seems particularly short and in need of expansion. Please incorporate some more background literature and detail. 

Line 38 - '...and genetic factors have been reported' - please provide more detail on these factors and their relationships with cognitive decline . E.g., 'physical' - what do you mean exactly? And what is the direction of association? 

Lines 39+ - you highlight that social norms have been proposed as a social determinant of health - can you provide some more detail? E.g., theoretical mechanism of associations or evidence linking social norms to health, in particaulr cognitive decline or factors identified as risk for cognitive decline. This also applies to you discussion of sekentei here.

Line 42 to 43 - if line 43 is a new paragraph please inset, if not please ensure it follows on from the previous statement.

Methods:

Line 72 - please include participation rate 

Line 82 - if possible, include reliability/validity informtion re the Mini-Mental State examination

Line 85 - the way you present your cut-points here is confusing (e.g., 23/24). How you present them in line 89 is much clearer. You also use this method of presentation in line 119.

Line 99 - please provide reliability/validity information in text beyond just stating that is has been confirmed.

Line 101 - why did you categorise? Please include a statement here.

Covariates - why did you choose to include these particular covariates? Please include a statement. Also, in this section it seems you also collected clinical measures from participants - information on this process should be presented in 'Study sample and Data collection'

Stats - line 126 - why did you choose to use multinomial regression as opposed to e.g., ordinal regression or even using the outcome as a continuous measure? Was there an a priori reason? Please include the reasoning for clarity.

Results

How does your sample compare to the target population?

Table 3 - please include the type of regression in the table title.

Why did you select quintile 4 as the reference category?

Daily walking time "longer" please define this in the table footer

Discussion line 215 - please correct the English in the sentence including "...because this was a cross-sectional study, a longitudinal study...". This is an important point but not currently clearly expressed.

Author Response

Reviewer #1

Thank you very much indeed for your careful review of our manuscript. We have replied to the reviewers’ comments in point-by-point manner below. In the manuscript, we made changes in response to the reviewers’ comments red-lettered.

  1. This section seems particularly short and in need of expansion. Please incorporate some more background literature and detail. 

Response: In accordance with the reviewer’s suggestion, we expanded the descriptions in the introduction section, particularly regarding social norms.

  1. Line 38 - '...and genetic factors have been reported' - please provide more detail on these factors and their relationships with cognitive decline. E.g., 'physical' - what do you mean exactly? And what is the direction of association? 

Response: We apologize for this oversight. We carefully checked the relevant papers (references #4 and #5) again, and found no factor related to “physical.” In the revised manuscript, we added some examples for each factor and also explained the direction of the association with regard to physical activity (lines 42–47).

“The risk factors for cognitive decline have been explored in many previous studies: medical (for example, diabetes and hypertension), nutritional (e.g., fruit and vegetable intake), socioeconomic (e.g., education and occupation), behavioral (e.g., smoking, physical activity, and cognitive engagement), and genetic (e.g., apolipoprotein E [APOE]) factors have been reported in systematic reviews [4,5]. For example, evidence for physical activity as a means to decrease risk of cognitive decline was shown both in observational studies and in a randomized controlled trial [4].” (lines 42–47)

  1. Lines 39+ - you highlight that social norms have been proposed as a social determinant of health - can you provide some more detail? E.g., theoretical mechanism of associations or evidence linking social norms to health, in particaulr cognitive decline or factors identified as risk for cognitive decline. This also applies to you discussion of sekentei here.

Response: In accordance with the reviewer’s advice, we added further information on social norms in the introduction section with appropriate references (lines 48-54).

“In discussions about the social determinants of health, the importance of socio-cultural factors such as social norms has been proposed [6]. Social norms are defined as unwritten rules about how to behave in a particular social group or culture [7]. People interpret the attitudes and behaviors of others around them as information about what should be done in a given situation, and often act accordingly [8]. Indeed, health behaviors can be influenced by social norms: physical exercise [9], smoking [10], and drug use [11]. However, to the best of our knowledge, no previous studies have examined the socio-cultural determinants of cognitive decline, including social norms.” (lines 48-54)

  1. Line 42 to 43 - if line 43 is a new paragraph please inset, if not please ensure it follows on from the previous statement.

Response: This was a new paragraph, and therefore, we inserted space.

  1. Line 72 - please include participation rate. 

Response: In accordance with this advice, we added the participation rate (527/1,346=39.2%; line 88).

“…527 people (participation rate: 39.2%)” (line 88)

  1. Line 82 - if possible, include reliability/validity information re the Mini-Mental State examination.

Response: Unfortunately, we did not assess participants’ cognitive function except for that captured by the MMSE-J. Thus, we can indicate neither the reliability nor the validity for our sample. However, the MMSE is used globally to assess cognitive function, and its reliability and validity have been shown in previous studies.

  1. Line 85 - the way you present your cut-points here is confusing (e.g., 23/24). How you present them in line 89 is much clearer. You also use this method of presentation in line 119.

Response: In accordance with this advice, we changed the presentation of the cutoff points (lines 100-103).

“This study adopted two cut-off points: ≤ 23 and ≤ 26. The former cut-off threshold is widely used to detect cognitive decline (a score of ≤ 23 indicated cognitive decline) [19,20]. The latter cut-off score has also been used in some studies to distinguish people with mild cognitive impairment (MCI) and healthy individuals (a score of ≤ 26 indicates MCI) [21,22].” (lines 100-103)

  1. Line 99 - please provide reliability/validity information in text beyond just stating that is has been confirmed.

Response: In accordance with the reviewer’s request, we explained the types of validity/reliability examined in previous related work (lines 111-113).

“The validity (i.e., content validity and construct validity) and reliability (i.e., internal consistency reliability) of this scale have been confirmed [16].” (lines 111-113)

  1. Line 101 - why did you categorise? Please include a statement here.

Response: Because no previous study had examined the relationship between sekentei and cognitive function, we were unsure whether the relation was linear. Therefore, we divided the sekentei score into some categories. In the revised manuscript, we explain this further (lines 114-117).

“Because there was no previous research examining the relationship between Sekentei Scores and cognitive function, it was unclear whether the relation was linear. Therefore, in the analysis, we divided the scores into quintiles (≤ 36, 37–39, 40–41, 42–44, and ≥ 45).” (lines 114-117)

  1. Covariates - why did you choose to include these particular covariates? Please include a statement. Also, in this section it seems you also collected clinical measures from participants - information on this process should be presented in 'Study sample and Data collection'

Response: We chose these covariates because they were considered to be confounders of the relation between sekentei and cognitive function. We added this point in the revised manuscript (lines 119-121). We also added descriptions to measure APOE genotype in the methods section (lines 137-141).

“Sociodemographic factors, health behaviors, health conditions, and genetic factors were used as covariates because these factors were considered to be confounders of the association between sekentei and cognitive function.” (lines 119-121)

To examine genetic factors, APOE genotyping was performed. Genomic DNA was obtained from whole-blood samples to determine APOE genotypes using a standard polymerase chain reaction technique. The technicians handling the coded DNA specimens were blinded to the diagnosis. The categorical variable of APOE was classified by genotype as ε2ε2, ε2ε3, ε2ε4, ε3ε3, ε3ε4, and ε4ε4 [26].” (lines 137-141)

  1. Stats - line 126 - why did you choose to use multinomial regression as opposed to e.g., ordinal regression or even using the outcome as a continuous measure? Was there an a priori reason? Please include the reasoning for clarity.

Response: As mentioned above, no study had tested the relation between sekentei and cognitive function. We thought that it would be beneficial to show the association between sekentei and MCI and between sekentei and cognitive decline, respectively, to understand the association in detail. Therefore, we used a multinomial regression model instead of ordinal regression and linear regression models. We added an explanation for this in the manuscript (lines 145-148). 

“Because we divided cognitive function into three categories (i.e., cognitive decline, MCI, and cognitively healthy), we adopted a multinomial logistic regression model to understand the detailed relationship between sekentei and cognitive function.” (lines 145-148)

  1. How does your sample compare to the target population?

Response: Both age and gender closely matched the target population; that is, mean age and % men were 73.5 years old and 47.3%, respectively, in this sample, and 73.6 years old and 47.0%, respectively, in the target population. In the revised manuscript, we added information for mean age and % men in the methods section (lines 83-84) and explained that these were similar to the target population (lines 157-158).

“Consequently, from a total of 15,792 eligible participants (average age 73.6, 47.0% men), …”(lines 83-84)

“The average age was 73.5 years old (standard deviation: 5.6) and 47.3% were men, which was similar to the characteristics of the target population.” (lines 157-158)

  1. Table 3 - please include the type of regression in the table title.

Response: We added the type of the regression analysis in the title of Table 3.

  1. Why did you select quintile 4 as the reference category?

Response: The average score of the Sekentei Scale (i.e., 41.6) was between the 3rd quintile and the 4th quintile, and its median was 42, which was included in the range of the 4th quintile. In addition, as shown in Table 3, the proportion of those with MCI/cognitive decline was the lowest in the 4th quintile. Therefore, in this analysis, we set this category as the reference. In the revised manuscript, we added information for the median of the Sekentei Scale (lines 162-163).

“The average Sekentei Scale score was 41.6 (standard deviation: 5.0, median: 42).” (lines 162-163)

  1. Daily walking time "longer" please define this in the table footer.

Response: In accordance with this request, we added an explanation in the footnote (Table 3).

  1. line 215 - please correct the English in the sentence including "...because this was a cross-sectional study, a longitudinal study...". This is an important point but not currently clearly expressed.

Response: In accordance with the reviewer’s advice, we rephrased this sentence in the revised manuscript (lines 238-240).

“Second, because of the cross-sectional nature of this study, causal relationships could not be determined; longitudinal data are required, which should be collected in a future study.” (lines 238-240)

Reviewer 2 Report

The current study examined the relationship between sekentei and cognitive function, using data from the NEIGE study, which consists of randomly sampled community-dwelling older Japanese people. The authors found that both higher and lower sekentei levels were negatively associated with lower cognitive function, particularly MCI, providing novel insights regarding how cognitive function may be related to sekentei tendencies, as this may also be related/similar to neuroticism personality traits. The current findings contribute to the understanding of the cultural determinants and behavioral aspects of cognitive decline.

Overall, I feel the paper is novel, well-written, the data are interesting and useful, and the work merits publication.  I don’t have any suggested modification, but wonder how this result may/may not apply to those over 85 (likely a select sample).  Does moderate levels of sekentei promote longevity after this age?   Can sekentei be modified at age 50 to promote healthy aging?  This could be communicated/discussed as an avenue for future research.

Author Response

Reviewer #2

Thank you very much indeed for your careful review of our manuscript. We have replied to the reviewers’ comments in point-by-point manner below. In the manuscript, we made changes in response to the reviewers’ comments red-lettered.

  1. Overall, I feel the paper is novel, well-written, the data are interesting and useful, and the work merits publication.  I don’t have any suggested modification, but wonder how this result may/may not apply to those over 85 (likely a select sample). Does moderate levels of sekentei promote longevity after this age? Can sekentei be modified at age 50 to promote healthy aging? This could be communicated/discussed as an avenue for future research.

Response: This is an important point. While we did not include those aged ≥ 85 years in the survey, and therefore cannot discuss the applicability of our findings to this age group, we added this point as a limitation (lines 250-252) in the conclusion. Furthermore, we think that it would not be easy to adjust one’s sense of sekentei. Therefore, it is important to consider people’s sekentei level in the community when developing strategies for prevention of cognitive decline.

“Finally, the participants in this study were 65–84 years old; therefore, further examination is necessary to determine whether our findings can be applied to other age groups, such as those ≥ 85 years old.” (lines 250-252)
